# First Detection of Tetrodotoxin in Bivalves and Gastropods from the French Mainland Coasts

**DOI:** 10.3390/toxins12090599

**Published:** 2020-09-16

**Authors:** Vincent Hort, Nathalie Arnich, Thierry Guérin, Gwenaëlle Lavison-Bompard, Marina Nicolas

**Affiliations:** 1Université Paris-Est, ANSES, Laboratory for Food Safety, F-94701 Maisons-Alfort, France; thierry.guerin@anses.fr (T.G.); gwenaelle.lavison-bompard@anses.fr (G.L.-B.); marina.nicolas@anses.fr (M.N.); 2ANSES, Risk Assessment Directorate, F-94701 Maisons-Alfort, France; nathalie.arnich@anses.fr

**Keywords:** tetrodotoxins (TTXs), bivalve molluscs, gastropods, hydrophilic interaction liquid chromatography associated with tandem mass spectrometry (HILIC-MS/MS), tetrodotoxin shellfish poisoning (TSP)

## Abstract

In 2015, tetrodotoxins (TTXs) were considered a potential threat in Europe since several studies had shown the presence of these toxins in European bivalve molluscs. In this study, we investigated the occurrence of TTXs in 127 bivalve samples (mussels and oysters) and in 66 gastropod samples (whelks) collected all along the French mainland coasts in 2017 and 2018. Analyses were carried out after optimization and in-house validation of a performing hydrophilic interaction liquid chromatography associated with tandem mass spectrometry (HILIC-MS/MS) method. The concentration set by European Food Safety Authority (EFSA) not expected to result in adverse effects (44 µg TTX equivalent/kg) was never exceeded, but TTX was detected in three mussel samples and one whelk sample (1.7–11.2 µg/kg). The tissue distribution of TTX in this whelk sample showed higher concentrations in the digestive gland, stomach and gonads (7.4 µg TTX/kg) than in the rest of the whelk tissues (below the limit of detection of 1.7 µg TTX/kg). This is the first study to report the detection of TTX in French molluscs.

## 1. Introduction

Tetrodotoxin (TTX) is a naturally occurring neurotoxin known to cause food poisoning in humans [1]. This toxin is a hydrophilic molecule (LogP = −6.2) whose molecular formula is C_11_H_17_N_3_O_8_ (m/z = 319.27 g/mol). Its structure consists of a guanidinium group bound to a highly oxygenated carbon skeleton, which has a 2,4-dioxaadamantane moiety containing five hydroxyl groups. To date, at least 30 structural analogues of TTX have been identified and belong to the TTX group (TTXs). Their degree of toxicity varies with their structure [2].

Historically, TTX intoxications have occurred mainly in Asia, and more specifically in Japan, after the consumption of certain species of fish belonging to the genus *Tetraodon* (e.g., globefish or fugu). A recent review indicates that the main source of intoxication worldwide is fish [3], involved in 59.9% of the cases (*n* = 1817), followed by gastropods (20.9%, *n* = 634), arthropods (16.2%, *n* = 492) and cephalopods (2.9%, *n* = 89). The fatality rate is significantly higher following the consumption of fish (16.3%) compared to other seafood products (gastropods, 4.9%; arthropods, 2.4% and cephalopods, 2.2%).

TTX had never been considered a potential threat in European marine waters, until the toxin was identified for the first time in 2008 as the causal agent of a food poisoning case linked to the consumption of *Charonia lampas* (gastropod) harvested in southern Portugal and purchased in Malaga, Spain [4,5]. The concentration measured in the meal leftovers (digestive glands) was 315.0 mg TTX/kg. A few years later, the presence of TTX and some of its analogues (4-epi-TTX, monodeoxy-TTX and 5,6,11-trideoxy-TTX) was confirmed in Portuguese samples of gastropods: *Monodonta lineata*, *Gibbula umbilicalis* and *Charonia lampas* [6].

More recently, in 2015, several European studies on the occurrence of TTX and its analogues in European bivalve molluscs were published. In the United Kingdom, the presence of TTX, 4-epi-TTX, 5,6,11-trideoxy-TTX and 4,9-anhydro-TTX was revealed in samples of bivalve molluscs (*Mytilus edulis* and *Crassostrea gigas*) collected between 2013 and 2014 in the English Channel [7]. A maximum level of 120 µg TTX/kg was measured in a sample of oysters. In Greece, several samples of bivalve molluscs (mussels and venus clams) collected between 2006 and 2014 were analysed with a maximum level of 203 µg TTX/kg quantified in the digestive glands of mussels [8]. Furthermore, this study showed the presence of TTX in European marine waters as early as 2006, based on samples of digestive glands in mussel samples, stored for nearly 10 years that contained a sum of TTXs ranging from 69 to 77 µg/kg. In the Netherlands, TTX was measured in whole mussel and oyster meat collected in 2015, with a maximum level of 124 µg/kg [9].

These new data raised many concerns in Europe and TTXs were quickly perceived as a potential emerging human health hazard. No regulatory limits had yet been set and this group of toxins was not monitored in shellfish. In the Netherlands, national sanitary measures were introduced and the European Commission called on the European Food Safety Authority (EFSA) Panel on Contaminants in the Food Chain to examine this issue [10]. In April 2017, EFSA published a scientific opinion on the risks associated with TTXs in marine bivalve molluscs and gastropods [9]. An acute reference dose (ARfD) of 0.25 µg/kg body weight (bw)/day for the sum of TTXs (i.e., TTX and its analogues) was derived. This dose is based on an oral mouse study that showed no adverse effects at 25 µg/kg bw. [11]. Considering a maximal shellfish flesh consumption of a 400 g for a 70 kg adult, EFSA considered that a concentration below 44.0 µg TTX equivalent (eq)/kg of shellfish meat would not result in adverse effects in humans. The opinion identified a series of uncertainty factors and several limitations due to insufficient data that led to the establishment of a number of recommendations to provide a more refined exposure assessment in the future. In particular, EFSA called for more data on the content of TTX and its analogues in the edible parts of bivalve molluscs and gastropods from European waters and more occurrence data from different Member States. 

Several European studies have been conducted since the EFSA risk assessment [12,13,14,15,16]. All these new data were obtained using hydrophilic interaction liquid chromatography associated with tandem mass spectrometry (HILIC-MS/MS). Although other detection methods exist, HILIC-MS/MS is currently the most suitable method for the identification and quantification of TTXs, and is recommended by EFSA for the acquisition of occurrence data [9,17,18]. HILIC-MS/MS offers excellent selectivity, sensitivity and robustness.

To date, the only French data came from a study conducted by the French Research Institute for Exploitation of the Sea (IFREMER) [9], in which 90 samples of digestive glands from oysters (*n* = 23) and mussels (*n* = 67) were analysed. All samples were below their limit of detection (LOD) of 33 µg/kg.

To respond to the need for occurrence data expressed by EFSA [9], an exhaustive national study was set up to obtain a better knowledge of the TTXs levels in French molluscs. In this study, sampling was designed to encompass spatial variability, with several sampling sites distributed all along the French mainland coasts (English Channel, Atlantic Ocean and Mediterranean Sea). Samples of bivalve molluscs (mussels and oysters) and gastropods (whelks) were included in this study. Temporal variability was also considered with a significant sampling period spanning 2017 and 2018, depending on the shellfish species (2017-2018 for whelks and 2018 for bivalve molluscs). An existing protocol was optimized [19,20,21] before analysing the samples. This optimized protocol resulted in a high-performance method, fully in-house validated and able to detect and quantify TTX and several of its analogues in shellfish at very low levels. This method was further optimized to obtain an even more sensitive variant, equally in-house validated.

## 2. Results

### 2.1. In-House Method Validation

Several adjustments were made to the procedure developed by Boundy et al. [19], and optimized by Turner et al. and the European Reference Laboratory (EURL) for marine biotoxins [20,21] 

First, the extract clean-up step was removed, because our tests showed similar recoveries with or without this step. With the clean-up step, mean recoveries were 103% for mussels, and 105% for oysters, and, respectively 103% and 111% without clean-up. The method precision was improved by removing this step because it adds a source of error that contributes to the method’s uncertainty.

A volume adjustment step was added. With the initial protocol, due to the non-adjustment of the extract volume, a difference in raw extract volumes was observed depending on the shellfish species and its water content. From several mussel samples, the mean extract volume observed was 6.4 mL, whereas for oysters a volume of 8.5 mL was measured. Not taking into account the accurate volume and considering in the calculations that the volume is 10 mL, led to an artificial improvement in the method’s trueness. Therefore, a volume adjustment step was added and the calculations took into account this accurate volume to obtain recoveries close to the true value.

Finally, a double extraction of the samples with 1% acetic acid was implemented before volume adjustment to 20 mL. This resulted in a significant improvement in recovery. With a simple extraction of mussel samples and an adjustment of the volume to 10 mL (with exact calculation of the results), the mean recovery was 61%, whereas it was 86% with a double extraction (5.0 mL of 1% acetic acid, followed by 10.0 mL of 1% acetic acid). 

This method was then fully in-house validated. First, matrix effects were studied for mussels, oysters and whelks. No significant effects were observed. For all samples and matrices, slope variations between matrix curves and solvent curves ranged from −5.7% to +11.6% (Table 1). The acceptance criterion was ±20%. Therefore, we opted for solvent calibration and not matrix-matched calibration curves. 

The linearity of calibration curves was studied after analysis of variance (ANOVA) tests and plotting the studentized residuals. Regardless of the weighting method applied, ANOVA was satisfactory, for the regression and for the lack of fit. However, the studentized residuals showed a lower number of outlier residuals with a 1/X weighting (*n* = 1 out of 24) than with a 1/X² weighting or no weighting (*n* = 3 in both cases). Therefore, 1/X weighting was applied for calibration throughout this study. 

Trueness and precision were very satisfactory and fulfilled the performance criteria defined in the CEN/TR 16059 standard [22]. For mussels and oysters, recoveries ranged from 93% to 111% (Table 2). The repeatability coefficient of variation, CVr, varied from 5.8% to 10.2% over the validity domain, and the intermediate precision coefficient of variation, CV_IP_, was between 7.2% and 10.2%. For whelks, recoveries were slightly lower, with values ranging from 74 to 80%. Precision was slightly higher, with repeatability ranging between 6.6% and 19.8% and between 12.2% and 19.8% for intermediate precision. Considering the satisfactory trueness of the method for the three matrices considered, we did not correct the results for the recoveries in this study.

The sensitivity of the method was also characterized. The limit of detection (LOD) was defined as the concentration that gives a signal-to-noise (S/N) ratio of 3:1 for at least one of the transitions followed (quantification and qualifier transitions), and was calculated from the lowest spiking concentration characterized. This LOD concentration was estimated at 3.8 µg/kg for mussels, oysters and whelks. The limit of quantification (LOQ), defined as the lowest validated level, was 15.0 µg/kg (experimental determination). 

### 2.2. Sample Analysis

All bivalve molluscs collected were analysed (*n* = 127). TTX was not quantified in any of the samples. The levels were below the LOD (3.8 µg/kg) in 98% of the samples (*n* = 124) and between the limit of detection and the limit of quantification (LOQ = 15.0 µg/kg) in 2% of the samples (*n* = 3) (Table 3). 

For gastropods, 66 samples were analysed. TTX could not be quantified in any of the samples. As for bivalves, TTX levels were below the LOD for 98% of the samples (*n* = 65) and between the LOD and the LOQ for one sample.

None of the TTX analogues screened for were detected either in bivalves or in gastropods.

### 2.3. Complementary Investigations

To increase the possibility of quantifying TTX in the four samples with levels below the LOQ but above the LOD, we developed a more sensitive variant of the method. The injection volume was multiplied by 5: the injection volume was 10 µL instead of 2 µL with the regular method.

#### 2.3.1. In-House Validation of the Sensitive Variant of the Method

The performances of the sensitive variant of the method were studied at the estimated LOQ of 5.0 µg TTX/kg. Trueness and precision were satisfactory for the three shellfish species studied (Table 4), and the performance criteria defined in the CEN/TR 16059 standard were met [22]. This estimated LOQ of 5.0 µg/kg was therefore experimentally confirmed. The LOD was estimated at 1.7 µg/kg for mussels, oysters and whelks, based on an S/N ratio of 3:1.

#### 2.3.2. Analysis of Samples Containing Tetrodotoxin with the Sensitive Variant of the Method

The sensitive variant of the method confirmed the presence of TTX in the four samples analysed (Table 3). Concentrations in the mussel 18 BM 139, 18 BM 154 samples and in the whelk 17 ET2M 773 sample were between the LOD (1.7 µg TTX/kg) and the LOQ (5.0 µg TTX/kg) whereas the mussel sample 18 BM 107, TTX was quantified at 11.2 µg/kg (Figure 1).

#### 2.3.3. Distribution of TTX in the Whelk Sample 17 ET2M 773

Whelk samples were divided in two subsamples: the gonads, digestive gland and stomach were pooled together into one subsample, and the remainder of the whelk tissues in the other subsample. We therefore investigated the distribution of TTX in these two subsamples of 17 ET2M 773. After analysis, 7.4 µg TTX/kg were found in the gonads, digestive gland and stomach, whereas TTX was below the LOD (below 1.7 µg TTX/kg) in the remainder tissues of this whelk sample.

## 3. Discussion

For bivalve molluscs, recoveries ranged from 99 to 108%. To the best of our knowledge, the trueness of this method is the best among all in-house-validated procedures [8,16,20,21,23,24]. For gastropods, recoveries ranged from 74 to 80%. Only those obtained using a much higher application domain (400 to 1200 µg/kg) are better [25]. For mussels and oysters, the repeatability and intermediate precision ranged respectively from 4 to 10%, and 7 to 10%. These values are the lowest observed among the literature [21,26]. For whelks, the repeatability ranged from 7 to 20% and the intermediate precision between 12 and 20%. Similar precision was obtained by Nzoughet et al. in gastropods [25]. The sensitivity obtained was better or of the same order as most other published studies [12,14,24,25,27], and considered totally fit for purpose, though lower than the sensitivities obtained by Leão et al. and Turner et al. [13,21]. LOQ values of the latter are respectively of 0.9 and 0.8 µg TTX/kg. Since for both these methods, the concentration factor associated with sample preparation was similar to our method, the difference in sensitivity is due to the performance of the instruments used. An inter-laboratory validation of the HILIC-MS/MS method for the analysis of toxins belonging to the saxitoxin group (PSP), and of TTX in bivalve molluscs, conducted by Turner et al., demonstrated the wide range of sensitivity of MS instrument models [23]. The system used in the present work was classified in the “medium sensitivity” group, whereas the instruments used the above-mentioned studies belonged to the “high sensitivity” group. Therefore, it could be interesting to evaluate the performances of the present method with “high sensitivity” instruments.

For the sensitive variant of our regular method, increasing the injection volume from 2 µL to 10 µL resulted—as expected—in a slight decrease in accuracy, likely due to matrix effects. For all the shellfish species studied, the repeatability of the variant ranged from 9 to 13%, the intermediate precision was in the range of 12 to 24% and trueness was quite similar (91 to 101%). The accuracy was, therefore, satisfactory. On the other hand, the larger injection volume greatly improved the sensitivity of the method. The LOQ was divided by a factor of three (Method LOQ = 15.0 µg TTX/kg vs sensitive variant LOQ = 5.0 µg TTX/kg) and the LOD was halved (Method LOD = 3.8 µg TTX/kg vs sensitive variant LOD = 1.7 µg TTX/kg). However, with this variant method, a slight shift in the TTXs retention time was observed between the standards prepared in solvent, and the extracts, due to the higher injection volume. To circumvent this issue, the retention time of the positive quality control (spiked blank matrix) was used as the reference retention time to check the retention time criterion in sample extracts.

The occurrence of TTX was observed for the first time in French mainland marine waters in 2018. The implementation of a very sensitive method led to the detection of TTX in three mussel samples and one whelk sample. TTX was quantified at a concentration level of 11.2 µg TTX/kg in only one mussel sample collected in July in the Ingril Lagoon (Mediterranean Sea coast).

European studies conducted on samples collected between 2008 and 2016 revealed that, in 92% of the mollusc samples, TTXs were neither detected nor quantified. Nevertheless, concentrations above the health hazard threshold of 44 µg TTX eq/kg set by EFSA [9], have been observed in several samples (*n* = 32). These values ranged from 47 to 253 µg TTX eq/kg, except the extreme values of 315 000 and 370 000 µg TTX eq/kg of *Charonia lampas* gastropod collected in Portugal [4,5,7,8,9,12,28]. Since 2017, TTX levels in molluscs appear to have decreased in general. In Spain and Portugal, several studies have since been conducted [2,13,14,16]; for bivalves and gastropods, the proportion of samples where TTX was detected was very low (0.13%) and the maximum level measured was in a cockle sample from the Galician “rías” with a quantified value of 2.3 µg TTX/kg. In addition to the Iberian locations, two gastropod samples from the Moroccan coast [2] revealed the presence of TTXs (maximum value: 20 µg TTX eq/kg). In the Netherlands, the maximum level observed in 2017 was 51 µg TTX eq/kg in an oyster sample [29,30]. This maximum value is below the highest reported values, observed in 2015 and 2016, of 124 and 253 µg TTX eq/kg respectively in two oyster samples. Furthermore, 2017 was the year with the lowest number of samples exceeding the European health hazard threshold in this country (*n* = 1). Italy is an exception: in 2017 and 2018, concentrations of 541 and 216 µg TTX/kg were respectively observed in two mussel samples (*Mytilus galloprovincialis*) from the same sampling site “Ficariol San Piero” located in the Marano Lagoon [31].

Therefore, the results obtained in the present study are in agreement with the data collected since 2017 in Europe. No assumption have been made, so far, to explain the decrease in TTX occurrence in European molluscs. This temporal variability raises an issue to be addressed. Better knowledge about the sources and critical factors leading to the accumulation of TTXs in marine bivalves and gastropods are needed as controversy still remains. [18]. TTXs may be produced by several bacterial species (e.g., *Acinetobacter*, *Alteromonas*, *Bacillus*, *Micrococcus*, *Pseudomonas*, *Vibrio*) [32,33]. On the other hand, the simultaneous presence of *Prorocentrum minimum* dinoflagellates and TTX in molluscs was observed in Greece [8]. In pufferfish, an endogenous origin is not excluded [34].

Noteworthily, in the present study, TTX was only detected in mussels among bivalve molluscs, whereas, based on the available data, EFSA highlighted that oysters were more contaminated than mussels [9]. The sampling design implemented in the present work may explain the non-detection of TTX in oysters: 82% of the bivalve mollusc samples were mussels; oysters were therefore relatively under-represented.

TTX was detected, but not quantified in one of the whelk samples analysed. Its concentration was comprised between the LOD (1.7 µg TTX/kg) and the LOQ (5.0 µg TTX/kg) of the sensitive variant of the method. For the 65 other gastropod samples, TTX was not detected. These results will be useful for risk assessment, because EFSA emphasized the need for more occurrence data for this class of shellfish to provide a more reliable exposure assessment [9]. This recommendation appears fully justified, because the only human intoxication reported in Europe from shellfish occurred after consumption of a gastropod species from Portugal [4]. The results obtained here did not show a significant difference between gastropods and bivalves in term of occurrence: TTX was detected in 2% of the samples in both mollusc groups.

TTX was detected in three mussels distributed all along the French mainland seaboards. One sample was from a Mediterranean lagoon (Ingril Lagoon), another was sampled in the Atlantic Ocean (Banc d’Arguin), and the last one was from the English Channel (Moulières d’Agon). Although the levels measured in bivalve molluscs were low in 2018, it would not be wise to extrapolate these results for the coming years.

In the United Kingdom (UK), intertidal or shallow waters (< 5 m depth) with lower levels of salinity in comparison to open marine waters, and temperatures above 15 °C were identified as presenting a greater level of risk of TTX occurrence in shellfish [35]. The bivalve sample from the Mediterranean Sea containing TTX was collected in spring (May), whereas the two others were sampled in summer (July). The Mediterranean climate (higher mean air temperature and amount of sunshine) and the shallow depth of the Ingril Lagoon (about 1 m) results in higher water temperatures than for the two other sampling sites. These environmental factors may explain the relatively early detection of TTX. In France, hydrological parameters and phytoplankton in coastal waters are monitored as part of an observation and monitoring programme [36]. Our bivalve sampling sites are associated with hydrological sampling sites, being identical or geographically close. During the three weeks before the sampling, the available data revealed that the water temperature was higher than 15 °C near the “Banc d’Arguin” and “Moulières d’Agon” sites. For the Ingril Lagoon, the water temperature when the mussel sample was collected was below 15 °C (11.9 °C); nevertheless, the water temperatures during the three previous weeks (*n* = 3) were higher than 15 °C (between 16.8 and 19.8 °C). Therefore, the depth and water temperature observed at these sampling sites are quite in compliance with those identified in UK.

For these three bivalve samples, the observed salinity at the hydrological sampling sites ranged from 32 to 37 g/L during the month before the sampling date. This range is in agreement with the salinity observed in other locations where TTX-positive samples were collected. In the Marano Lagoon in Italy, the measured salinity was 33 g/L, whereas, in the Galician “rías” in Spain, salinity was 35 g/L [13,31].

In Greece, the simultaneous presence of the *P. minimum* (also called *Prorocentrum cordatum*) and TTX in shellfish has been reported [8], with cell counts of up to 1890 cells/L. The production of TTX by this dinoflagellate is therefore a hypothesis to be investigated. We thus retrospectively checked the *P. minimum* cell concentrations in 2018 for the three locations where TTX was detected. Unfortunately, in 2018, *P. minimum* was not monitored close to “Ingril Sud” and “Moulières d’Agon”. However, data were available from two sites near “Banc d’Arguin”: “Arcachon - Bouée 7” and “Teychan bis” [36]. At “Arcachon - Bouée 7”, five weeks after the collection of 18 BM 139 (July), 600 cells/L were measured. At “Teychan bis”, four weeks before and three weeks after sampling, 1000 cells/L were counted. Although these values are similar to those observed in Greece [8], it is nevertheless difficult to correlate the presence of *P. minimum* with the TTX content in molluscs. During other periods in 2018, *P. minimum* was detected at higher concentrations than the levels measured at the period corresponding to the collection of sample 18 BM 139, but TTX was not detected in the samples collected at those times. For example, at both sites, the maximum values of *P. minimum* were observed three months earlier (April). At “Arcachon - Bouée 7”, 1400 cells/L were measured and at “Teychan bis” 2600 cells/L were observed. The TTX concentration in the shellfish sample collected during that month was below the method LOD of 3.8 µg TTX/kg. Moreover, *P. minimum* data were also available in 2018 for two of the locations studied: “Marseillan” and “Diana” points. For both sites, higher levels than in “Arcachon - Bouée 7” and “Teychan bis” were observed whereas TTX was not detected in bivalves collected all along 2018: up to 7000 cells/L for “Marseillan” point in August and up to 2600 cells/L for “Diana” point in April. Further studies are therefore required to investigate the correlation between the presence of *P. minimum* and the TTX content in bivalves.

The gastropod sample with a TTX concentration between LOD and LOQ (17 ET2M 773) was sampled in November 2017 in a natural whelk bed situated in the English Channel (west of the Cotentin Peninsula), close to Rimains Island near the town of Cancale. Comparing the GPS data with French coastal maps allowed us to estimate the mean water depth of this site [37]. Depths vary greatly depending on the tidal coefficients (between 3.5 and 17 m). The mean depth was estimated at about 10 m at that sampling site. Sampling was generally carried out at depth greater than 15 m and sometimes at more than 40 m. Among all the whelk samples, the 17 ET2M 773 sample was collected at the greatest depth as was sample 18 ET2M 137 (TTX result below the LOD of the method). Unlike the positive bivalve molluscs, the 17 ET2M 773 whelk sample was not carried out during spring or summer, but in autumn (November 2017). One week before sampling, the water temperature measured at the closest hydrological sampling point (“Mont St-Michel”, 5 km away) was 11.4°C at 1 m depth, and 14.5°C, three weeks prior. Therefore, the depth at the sampling point suggests that the water temperature was lower than 15°C for several weeks. However, with only one positive sample for gastropods, it is not possible to identify trends about the environmental parameters leading to the accumulation of TTX.

For sample 17 ET2M 773, higher concentrations were found in the gonads, digestive gland and stomach subsample (7.4 µg TTX/kg) than in the rest of the tissues (below LOD of 1.7 µg/kg). Therefore, TTX was at least four times more concentrated in the gonads, digestive gland and stomach than in the rest of the whelk. Reasoning in terms of quantities rather than concentrations helps to better understand this distribution. The quantities of the two whelk subsamples were weighed and were not equal. The subsample corresponding to the remaining tissues represented 76% of the weight of whelk tissues. Therefore, for 100 g of whole flesh of 17 ET2M 773, 80 ng of TTX were in the gonads, digestive gland and stomach and less than 57 ng were found in the rest of the whelk tissues. This quantity could not be determined precisely, because it was below the LOD of the sensitive method variant. We nevertheless demonstrated that a higher quantity of TTX was found in the gonads, digestive gland and stomach than in the rest of the whelk.

To date, there is little information on TTX distribution in European mollusc tissues [9]. For bivalve molluscs, the only available data are from Greece [8]. In one mussel sample, similar levels were measured in the digestive gland and the whole flesh (202.9 and 179.1 µg TTX/kg respectively). In New Zealand, the distribution of TTX was investigated in an endemic bivalve mollusc species (*Paphies australis*) [38,39], revealing that the siphon of this marine organism is the most contaminated organ. For gastropods, several Asian studies have been conducted on different species (*Natica lineata*, *Natica*, *vitellus*, *Polinices didyma*, *Polinices tumidus*, *Nassarius conoidalis*, *Nassarius glans*), but never from *Buccinum undatum* [18,40,41,42,43,44,45]. Except for *N. conoidalis*, all those studies revealed higher concentrations in the muscle than in the digestive gland of the gastropods. However, significant variability between specimens was observed with regard to TTX levels in the digestive gland, which were sometimes higher than in the muscle. In the absence of published data on TTX distribution in *B. undatum*, our results constitute the first report of TTX distribution in this species and call for confirmation with studies involving a high number of naturally contaminated samples.

## 4. Conclusions

The aim of this study was to investigate the occurrence of TTXs in bivalve molluscs and gastropods in mainland France. The analytical approach developed resulted in a high-performance, in-house-validated method. This is the first published report of TTX in French molluscs collected in 2017 and 2018 from several sampling sites spanning the entire mainland coastline: the Atlantic Ocean, the Mediterranean Sea and the English Channel. Although the European health hazard threshold of 44 µg TTX eq/kg was never exceeded, sample analysis revealed the presence of TTX in three bivalve and one gastropod samples. Our results are consistent with those published in Europe since 2017, with a trend for a decrease in the occurrence of TTXs. However, this does not presage the situation for the coming years. Since 2019, the French scheme for the monitoring of emerging toxins (EMERGTOX) continues to survey TTXs in bivalves in 11 mainland coastal areas. This study also provides TTXs data for gastropods. These results will be useful considering the lack of European data for this class of molluscs. This study also provided the first data on TTX distribution in whelks and should stimulate further investigation.

This study acted on several EFSA recommendations [9]. New European data on the occurrence of TTX and its analogues in bivalve molluscs and gastropods are now available. Our data for France complement those obtained in several neighbouring countries and thus enhance European databases.

## 5. Materials and Methods

### 5.1. Chemicals and Reagents

All solutions were prepared with liquid chromatography-mass spectrometry (LC-MS) grade chemicals and ultrapure water (18.2 MΩ cm) obtained by purifying distilled water with a Milli-Q system associated with an Elix 5 pre-system (Millipore S.A., St Quentin-en-Yvelines, France). The TTX standard was a certified reference material purchased from Cifga Laboratory© (Lugo, Spain). From this standard, a stock solution of TTX (1.0 µg/mL) was prepared in 0.03 M acetic acid. This solution was used to spike the blank samples used as quality control and for method validation. A working standard solution (125 ng/mL) was prepared in acetonitrile/water/acetic acid 80:20:0.25 (*v*/*v*/*v*) from the stock solution. This working standard solution was diluted with acetonitrile/water/acetic acid 80:20:0.25 (*v*/*v*/*v*) for the preparation of the calibration levels. The following TTX concentrations were obtained: 0.375, 1.250, 1.875 and 2.500 ng/mL. Acetonitrile, methanol and acetic acid and formic acid were purchased from Fisher Scientific (Loughborough, UK). Ultra-pure-grade carrier argon (Ar, 99.9999% pure) and nitrogen (N2, 99.999% pure) were purchased from Linde Gas (Montereau-Fault-Yonne, France).

### 5.2. Sampling Design

Bivalve molluscs were sampled all along the French mainland coasts (Figure 2). For bivalve molluscs, mussels (*Mytilus edulis* and *Mytilus galloprovincialis*) and oysters (*Crassostrea gigas*) were obtained from 11 sampling sites situated in the Atlantic Ocean, the English Channel and the Mediterranean Sea. The bivalve mollusc species studied are those mainly exploited in France. Throughout 2018, 1 kg of bivalve molluscs was sampled monthly. Nine sites were sampled for mussels and two for oysters. These sites are mainly located in active production areas. Overall, this sampling design included 132 bivalve mollusc samples, however, 127 were actually collected. The sample corresponding to the “Diana” site in the Mediterranean Sea could not be obtained from August to October, because the shellfish had been transferred elsewhere by the professionals working in this area, due to an increase in water temperatures. The December sample for the “Le Scoré” site could not be sampled either, due to unfavourable weather conditions. Finally, no samples were collected at the “Marseillan” site in September and December. During summer 2018, a high mortality event occurred in the Thau Lagoon where “Marseillan” is located. Losses reached 100% for mussels and 52% for oysters.

Between 2017 and 2018, 66 gastropod samples (*Buccinum undatum*) were collected from the three main French natural beds situated in the English Channel. These whelk beds were situated to the west of the Cotentin Peninsula, in the Bay of Seine and close to Cap Gris-Nez (Figure 2). Each sample was composed of 2 kg of individuals.

### 5.3. Sample Preparation

Bivalve molluscs were prepared as described in the EN 14526 standard [46]. Gastropods were initially sampled for heavy metal analyses and required separating whelk tissues into two subsamples. The first subsample came from one extremity of the whelk consisting of the gonad, the digestive gland and the stomach, and the other subsample was made up of the remaining whelk tissues. After sampling, whole whelks were thoroughly cleaned with tap water to remove sand and foreign material. The shell of each whelk was then carefully broken by using a hammer. Tissues were removed and divided into two parts as described above. Both subsamples were drained for 5 min in a sieve and weighed. Whelk tissues were then homogenized using a Grindomix GM200 grinder (Retsch GmbH, Haan, Germany), and at least 100 g was obtained. For TTXs analysis, whole flesh test portions were reconstituted based on the mass proportions of each whelk subsample. Homogenized bivalve molluscs and gastropods tissues were stored at −20 °C.

### 5.4. Tetrodotoxin Extraction

The TTXs extraction method was based on the procedure developed by Boundy et al. [19], and optimized by Turner et al. and the EURL for marine biotoxins [20,21]. We refined this protocol here, and several modifications were implemented (Appendix A).

A 5.0 ± 0.1 g portion of homogenized shellfish tissue was weighed into a 50 mL centrifuge tube. Then, 5.0 mL of 1% acetic acid was added and the sample was mixed by vortexing for 90 s. The sealed tube containing the sample was placed in a boiling water bath for 5 min and then in a cold-water bath for 5 min. The sample was vortexed for 90 s again, before being centrifuged 10 min at 9000× g. The supernatant was collected in a 50 mL centrifuge tube. The test portion was then extracted again with 10.0 mL of 1% acetic acid, vortexed for 90 s for the third time and centrifuged 10 min at 9000× g. The supernatant was transferred to a 50 mL centrifuge tube. The volume of this crude extract was adjusted to 20 mL with 1% acetic acid. This extract was 1:10 diluted with water/acetonitrile/acetic acid 88.9:11.1:0.167 (*v*/*v*/*v*) and filtered on a 0.2 µm nylon (Xtra PA) syringe filter (Chromafil™, Macherey-Nagel GmbH & Co. KG, Düren, Germany).

### 5.5. Liquid Chromatography Conditions

The LC system was an Accela 1250 (Thermo Fisher Scientific, San Jose, CA, USA). Ultra-high-performance liquid chromatographic (UHPLC) separation of TTXs was performed using an Acquity UPLC BEH Amide column or an Acquity UPLC BEH Glycan column (Waters, Milford, MA, USA). Both columns had similar dimensions (150 × 2.1 mm, 1.7 μm particle sizes, 130 Å) and each was equipped with its specific Vanguard pre-column system (5 × 2.1 mm, 1.7 µm particle sizes, 130 Å). The column temperature was set to 70°C. Eluent A was composed of water/formic acid/ammonia 250:0.0375:0.15 (*v*/*v*/*v*) and eluent B of acetonitrile/water/formic acid 700:300:0.1 (*v*/*v*/*v*). The TTX separation gradient was programmed as follows: 98% B at 0.4 mL/min (hold 4 min), 98–50% B with the same flow rate (in 3.5 min), 50% B with a linear increase in the flow rate to 0.6 mL/min (hold 1.5 min), 50–98% B with a linear increase in the flow rate to 0.8 mL/min (in 0.5 min), 98% B with the same flow rate (hold 0.6 min). This gradient enables the separation of all the screened TTX analogues. A chromatogram obtained after injection of a standard containing most of these analogues is given in Figure 3. Additional chromatographic methods were necessary for column cleaning, equilibration and storage. These methods are fully described in references [19,20,23,47]. In the present study, 2 µL were injected into the system. A more sensitive variant of this method required a 10 µL injection volume.

### 5.6. Tandem Mass Spectrometry Conditions

TTXs detection was performed with a TSQ Vantage triple quadrupole mass spectrometer (Thermo Fisher Scientific), equipped with an electrospray ionization (ESI) source (HESI-II probe). The mass spectrometer was operated in selected reaction monitoring (SRM) mode. The spray voltage was 3500 V in positive ionization mode. The source temperature was set at 350 °C and capillary temperature at 300 °C. Nitrogen was used as the nebulizing gas with a sheath gas pressure of 40 (arbitrary units) and an auxiliary gas pressure of 15 (arbitrary units). The collision gas was argon, with a gas pressure of 1.5 mTorr. For all compounds, the S-lens voltage was set to 120 V. One transition was used for quantification (Q) and another as a qualifier transition (q). The optimized compound-dependent parameters are listed in Table 5. In addition to TTX and its analogues, transitions for arginine (Arg) and hydroxy-arginine (OH-Arg) were incorporated in the method and monitored. These amino acids suppress the TTX response in MS [21]. A mass resolution of 0.7 Da (full width at half maximum) was set for the first and the third quadrupoles (Q1 and Q3). Instrument control and acquired data were handled by a computer equipped with TSQ Tune Master version 2.6.0, Xcalibur version 4.1.31.9 and TraceFinder™ version 4.1—EFS (Thermo Fisher Scientific).

### 5.7. Method Validation Methodology

The method was fully in-house validated for TTX. Matrix effects were studied for mussels, oysters and whelks. For each matrix, four calibration curves were prepared from blank extracts obtained after the extraction of four different samples. Each calibration curve was composed of four concentrations: 0.375. 1.275. 1.875 and 2.500 ng/mL. An extract volume of 90%was maintained in all vials injected. Matrix effects were calculated by comparison of the slopes of the matrix curves with the slopes of the solvent curves, using the equation:(1)$$\mathrm{Matrix}\;\mathrm{effect}\;\left(\%\right)\;=\;\frac{\mathrm{matrix}\;\mathrm{curve}\;\mathrm{slope}-\;\mathrm{solvent}\;\mathrm{curve}\;\mathrm{slope}}{\mathrm{solvent}\;\mathrm{curve}\;\mathrm{slope}}$$

The linearity of the regression model was statistically evaluated. Five repetitions of the calibration curve were prepared and injected in the same sequence. A Fisher-Snedecor test with a significance level of α = 0.01, and the studentized residuals were plotted. Three weighting methods were tested: no weighting, 1/X weighting and 1/X^2^ weighting.

The accuracy of the method was also characterized for the three matrices studied: mussels, oysters and whelks. Each of the three experimental designs included four concentration levels (15.0; 20.0; 44.0 and 100.0 μg TTX/kg), four series repeated on different days. Spiked samples followed the whole analytical procedure, including extraction. For each series, two replicates of each of the four concentration levels were analysed. The two HILIC column references were used. Two series were done using a BEH Amide column and the two others using a BEH Glycan column. For each series, a different blank sample was used for spiking to take into account between-sample variability. Performance criteria established to validate the accuracy and the sensitivity of the method were those described in the CEN/TR 16059 [22].

The LOD was defined as the concentration that gives an S/N ratio of 3:1 for the least intense of the two transitions that we monitored (quantitative and qualifier), and was calculated from the lowest concentration characterized. The LOQ was defined as the lowest concentration level experimentally validated in terms of accuracy.

Considering the analytical strategy implemented, the variant of the method was only characterized at the estimated LOQ (5.0 µg TTX/kg) with experimental designs carried out the same way as for the regular method. LOD and LOQ were also determined with the approaches previously described. Performances of this method variant also had to fulfil the criteria described in the CEN/TR 16059 standard [22].

### 5.8. Quality Control

To ensure reliable results, samples were analysed in sequences including several internal quality controls (IQCs). When acceptance criteria were not all met, the results were discarded and samples were re-analysed. One of these IQC was a blank matrix, analysed in the same conditions as the samples. To check for the absence of any contamination, the S/N ratio had to be below 3:1 for at least one of the two transitions monitored. Another IQC was a blank matrix spiked with TTX at the health hazard threshold (44 μg TTX equivalent/kg), and analysed in the same conditions as the samples. Depending on the sequence of analysis, the blank matrix was a mussel, an oyster or a whelk sample. To be accepted, results had to be within ± 2 times the standard deviation calculated using the Horwitz-Thompson estimator (relative standard deviation of 22%) (Appendix A). TTX was quantified using calibration curves injected at the beginning and the end of the sequence to establish the response function. The determination coefficient of the calibration curve had to be ≥0.98. The relative residuals were also calculated for each standard injected, and had to be lower or equal to 25%. Finally, the variation of the retention time for each sample had to be below 5% relative to the standard retention time or the spiked sample.

## Figures and Tables

**Figure 1 toxins-12-00599-f001:**
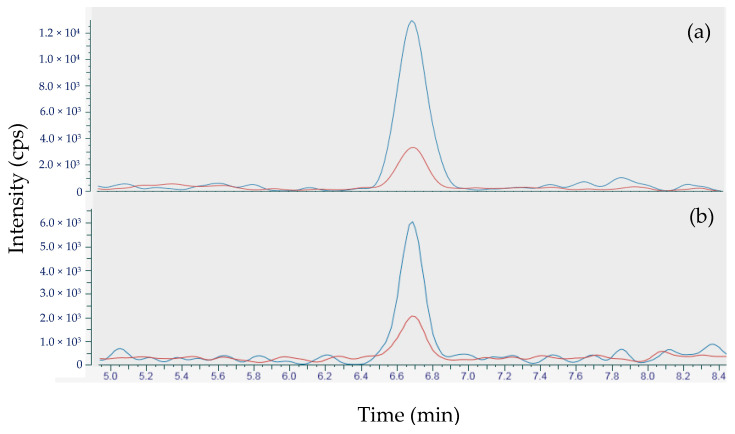
Tetrodotoxin (TTX) peaks obtained after analysis using the sensitive variant of the method, (**a**) for sample 18 BM 107 and (**b**) the positive control sample (5.0 µg TTX/kg) integrated in the same sequence (red line for the quantitative transition and blue line for the qualifier transition).

**Figure 2 toxins-12-00599-f002:**
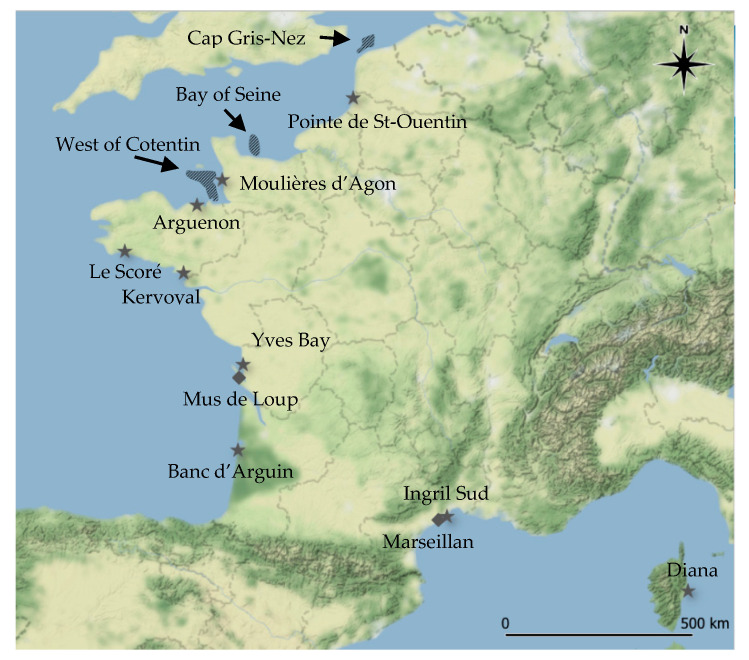
Locations of the bivalve mollusc sampling sites (
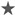
: mussels; 
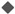
: oysters) and gastropod beds (
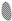
: whelks). Map generated using QGIS software (version 3.8.3) and map tiles by Stamen Design, under CC BY 3.0. Data by OpenStreetMap, under ODbL.

**Figure 3 toxins-12-00599-f003:**
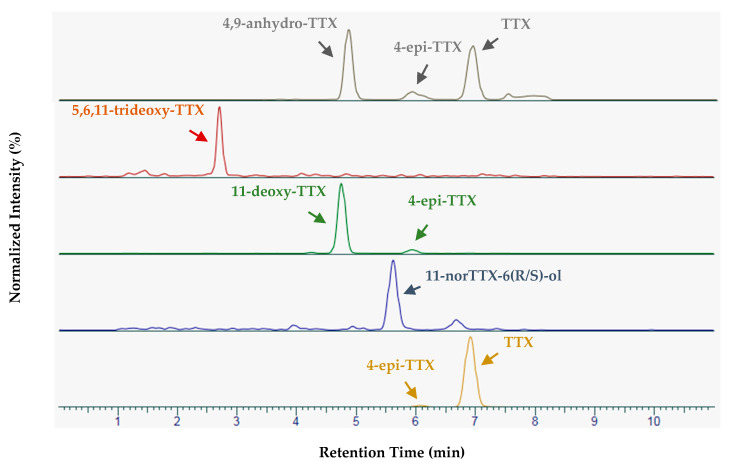
Separation obtained after injection of a standard solution containing tetrodotoxin (TTX) and some of its analogues.

**Table 1 toxins-12-00599-t001:** Tetrodotoxin (TTX) matrix effects expressed as the relative difference between the slopes of the matrix curves and the solvent curves.

Serial Number	Mussels	Oysters	Whelks
1	+6.0%	+8.9%	+11.6%
2	+5.2%	+0.7%	+4.5%
3	+6.7%	−2.8%	+4.9%
4	−0.8%	−5.7%	+3.3%
Mean	+4.3%	+0.3%	+6.1%

**Table 2 toxins-12-00599-t002:** Method performances for tetrodotoxin (TTX).

Shellfish	Method Characteristic	TTX Concentration (µg/kg)			
		15.0	20.0	44.0	100.0
Mussels	Recovery (%)	101	98	97	93
	Repeatability (CV_r_, %)	6.7	9.1	9.0	5.8
	Intermediate precision (CV_IP_, %)	8.3	9.1	9.0	7.2
Oysters	Recovery (%)	111	101	101	99
	Repeatability (CV_r_, %)	10.2	7.2	7.3	3.6
	Intermediate precision (CV_IP_, %)	10.2	7.2	8.7	7.7
Whelks	Recovery (%)	79	79	80	74
	Repeatability (CV_r_, %)	19.8	10.7	6.6	10.9
	Intermediate precision (CV_IP_, %)	19.8	15.1	13.9	12.2

**Table 3 toxins-12-00599-t003:** Shellfish samples with tetrodotoxins (TTXs) detected using the in-house validated analytical method and its sensitive variant.

Sample Number	Shellfish	Seaboard	Sampling Site	Sampling Month	Results	
					Method	Sensitive Variant
18 BM 107	Mussels (Bivalves)	Mediterranean Sea	Ingril Lagoon	May	< LOQ ^1^	11.2 µg TTX/kg
18 BM 139	Mussels (Bivalves)	Atlantic Ocean	Banc d’Arguin	July	< LOQ ^1^	< LOQ ^2^
18 BM 154	Mussels (Bivalves)	English Channel	Moulières d’Agon	July	< LOQ ^1^	< LOQ ^2^
17 ET2M 773	Whelks (Gastropods)	English Channel	West of Cotentin	November	< LOQ ^1^	< LOQ ^2^

^1^ LOQ of the method: 15.0 µg TTX/kg; ^2^ LOQ of the sensitive variant of the method: 5.0 µg TTX/kg.

**Table 4 toxins-12-00599-t004:** Analytical performance of the sensitive variant of the method at a tetrodotoxin (TTX) concentration of 5.0 µg/kg.

Method Characteristic	Mussels	Oysters	Whelks
Recovery (%)	91	100	93
Repeatability (CVr, %)	8.5	9.9	12.9
Intermediate precision (CVIP, %)	12.0	23.0	24.0

**Table 5 toxins-12-00599-t005:** Compound-dependent tandem mass spectrometry parameters for tetrodotoxin (TTX) and the screened analogues.

Compounds	Precursor Ion (m/z)	Collision Energy (V)	Product Ion ^1^ (m/z)
TTX/4-epi-TTX	320.1	23	302.1(Q)
		35	162.1 (q)
5,6,11-trideoxy-TTX	272.1	30	254.1 (Q)
		35	162.1 (q)
11-norTTX-6(R/S)-ol	290.1	30	272.1 (Q)
		35	162.1 (q)
4,9-anhydro-TTX	302.0	24	256.0 (Q)
		35	162.1 (q)
5-deoxy-TTX/11-deoxy-TTX	304.1	30	286.1 (Q)
		30	176.0 (q)
Arginine	175.0	30	70.0 (Q)
		30	60.0 (q)
Hydroxy arginine	191.0	30	86.0 (Q)
		30	68.0 (q)

^1^ Q: Quantitative transition; q: qualifier transition.

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
