# Peer review of "First Detection of Tetrodotoxin in Bivalves and Gastropods from the French Mainland Coasts"

_toxins, 2020, doi:10.3390/toxins12090599_

Round 1
Reviewer 1 Report
This manuscript reveal for the first time the presence of TTX in 3 bivalves and one gastropod collected on French coasts. Although I am not a specialist at all, the methods used seem to be robust, and the authors proposed several optimizations that will certainly prove to be useful. It is well-written, clear and easy to follow. The results are correctly interpreted. I have only a few minor comments, the (relatively) most important of them being the fact that the discussion falls a little bit short on explanations about the results obtained.
First, where does this TTX come from? There is a short paragraph about this in the discussion (L223-229), but nothing is said about what could explain the temporal variability in TTX in Europe (before 2010 vs 2010-2017 vs after 2017). Is there any hypotheses proposed in other articles that would explain how TTX prevalence can vary among years?
Then, several environmental parameters are explored (salinity, temperature, depth), but the authors do not clearly conclude about these environmental parameters: it seems that none of them are correlated with the presence of TTX in the four samples (but with such a low number of samples with TTX, one should not expect to have significant trends), but maybe other hypothesis have been proposed in the literature to explain such pattern. Indeed, I guess that if the authors looked at salinity, temperature and depth, it’s because these parameters have been looked at in the past: what was the results obtained in the past studies? And how does it relate to the results obtained here?
Again, these are only minor comments (see also the list below), and I would recommend to accept this manuscript for publication with minor modifications.
L79: it is not clear here wat this LOD is related to? To the detection method they used I guess?
L359: Maybe briefly explain why these 4 species (easy to collect, edible, known to have TTX?)
L365-366: “three main” instead of “main three”
L369: I can’t see the grey shapes in my pdf, but it is maybe just a problem of conversion.
L132: remove “be”
L163: why not testing this in the mussel 18 BM 107, which has the highest level of TTX?
L187: So maybe it could be suggested that the method designed here should be tested with “high sensitivity” instruments to be compared with results obtained by Leão et al. and Turner et al.
L221: “This site may be a 221 hotspot.” This is quite useless, except if it is completed (hotspot of what? What could be its origin? What would be the consequence?).
L270-285: and what about P. minimum concentration at other sites where TTX was not detected?
Author Response
Reviewer 1:
This manuscript reveal for the first time the presence of TTX in 3 bivalves and one gastropod collected on French coasts. Although I am not a specialist at all, the methods used seem to be robust, and the authors proposed several optimizations that will certainly prove to be useful. It is well-written, clear and easy to follow. The results are correctly interpreted. I have only a few minor comments, the (relatively) most important of them being the fact that the discussion falls a little bit short on explanations about the results obtained.
Answer: Thank you for your positive feedback. The point about the discussion has been taken into account through the following comments.
First, where does this TTX come from? There is a short paragraph about this in the discussion (L223-229), but nothing is said about what could explain the temporal variability in TTX in Europe (before 2010 vs 2010-2017 vs after 2017). Is there any hypotheses proposed in other articles that would explain how TTX prevalence can vary among years?
Answer: regarding the first question, the reference mentioned in the paragraph (review article, [17]), explains that: “to manage the risk associated for TTX in edible seafood, it is important to understand its origin, the mechanism in which the toxin enters the food supply and how the concentrations observed are accumulated. For TTX, this is difficult as controversy remains regarding its origin”. Others references mentioned in this review are in agreement with this conclusion. The sentence L248 was modified to highlight the existence of a controversy on this point.
Regarding the second question, no assumption have been made to explain the temporal variability of TTX in molluscs, so far. This point was also added in the main text (L247).
Then, several environmental parameters are explored (salinity, temperature, depth), but the authors do not clearly conclude about these environmental parameters: it seems that none of them are correlated with the presence of TTX in the four samples (but with such a low number of samples with TTX, one should not expect to have significant trends), but maybe other hypothesis have been proposed in the literature to explain such pattern. Indeed, i guess that if the authors looked at salinity, temperature and depth, it’s because these parameters have been looked at in the past: what was the results obtained in the past studies? And how does it relate to the results obtained here?
Answer: sentences have been added in the section 3. (L287, L330) to clarify our conclusion about the depth and the water temperature observed during sampling of the positive bivalve mollusc and gastropod samples.
Effectively, we looked at the salinity, temperature and depth because these parameters have been looked in the previous study of Turner et al., 2017. L274, we explained that “in the UK, intertidal or shallow waters (< 5 m depth) with water temperatures above 15°C were identified as presenting a greater level of risk of TTX occurrence in shellfish [20]”. This sentence L274 was modified to also integrate the conclusion about salinity of Turner study.
We also identified that the previous reference [20] was not correct, it was therefore not possible for the reviewer to understand this sentence by reading the publication mentioned. We are sorry for this mistake. The study initially mentioned was another publication of the same year and the same author. This reference [20] was therefore replaced by the reference [35]:
Turner, A.D.; Dhanji-Rapkova, M.; Coates, L.; Bickerstaff, L.; Milligan, S.; O’Neill, A.; Faulkner, D.; McEneny, H.; Baker-Austin, C.; Lees, D.N.; et al. Detection of Tetrodotoxin Shellfish Poisoning (TSP) toxins and causative factors in bivalve molluscs from the UK. Mar. Drugs 2017, 15, doi:10.3390/md15090277.
Again, these are only minor comments (see also the list below), and I would recommend to accept this manuscript for publication with minor modifications.
Answer: Thank you.
L79: it is not clear here wat this LOD is related to? To the detection method they used I guess?
Answer: Yes, it is the LOD of their method. We clarified the sentence L81.
L359: maybe briefly explain why these 4 species (easy to collect, edible, known to have TTX?)
Answer: We added a sentence L394 to explain the choice of the bivalve mollusc species studied. In fact, the bivalve mollusc species studied are those mainly exploited in France. For gastropods, as explained L375, the three main French natural beds are composed of Buccinum undatum so this specie was studied.
L365-366: “three main” instead of “main three”
Answer: Thank you. Done.
L369: I can’t see the grey shapes in my pdf, but it is maybe just a problem of conversion.
Answer: it seems that the shapes disappeared during the generation of the pdf because there were present in the word version. We will pay attention next time to correct this point during the pdf generation.
L132: remove “be”
Answer: Thank you. Done.
L163: why not testing this in the mussel 18 BM 107, which has the highest level of TTX?
Answer: It would have been very interesting, however only the gastropods were initially divided in two subsamples. All bivalve molluscs were prepared as described in the EN 14526 standard (L372). An homogenate of the whole flesh was prepared therefore it was not possible to analyse the tissue distribution of TTX for bivalve molluscs
L187: so maybe it could be suggested that the method designed here should be tested with “high sensitivity” instruments to be compared with results obtained by Leão et al. and Turner et al.
Answer: We fully agree with this suggestion. A sentence was added L211.
L221: “this site may be a 221 hotspot.” This is quite useless, except if it is completed (hotspot of what? What could be its origin? What would be the consequence?).
Answer: This sentence was removed.
L270-285: and what about p. Minimum concentration at other sites where TTX was not detected?
Answer: Several sentences have been added in the discussion L311 regarding the other sites. In fact, P. minimum data were also available in 2018 for two of the sites studied: “Marseillan” and “Diana” points. For both locations, higher levels than in "Arcachon - Bouée 7” and "Teychan bis" were observed whereas TTX was not detected in bivalves collected all along 2018: up to 7000 cells/L for “Marseillan” point in August and up to 2600 cells/L for “Diana” point in April.
Reviewer 2 Report
The manuscript “First detection of tetrodotoxin in bivalves and 2 gastropods from the French mainland coasts” deal with the detection of tetrodotoxin in 4 samples (127 collected) of shellfishes from the French coasts. Tetrodotoxin were quantified in only one sample. Tetrodotoxin is commonly associated to the consumption of contaminated fish in China and Japan, but some intoxications have occurred also in Europe. This is the first detection of tetrodotoxin in France. The authors also optimized the analytical method for the detection of tetrodotoxin using HILIC-MS/MS.
I found this work interesting and it can be accepted after minor changes.
Abstract - The abstract should be rewritten in order to be more concise and relevant. The first lines on tetrodotoxins are not necessary, especially because they detected only the founder compound of this group named tetrodotoxin.
Line 30- Please add references.
Line 95- considering that the method validation reported in Appendix A is not complex and long, I suggest removing the appendix A and adding all the information in the main text.
Lines 124-131- is this necessary in the results section? I suggest reporting only the actual number of samples and the sampling sites.
Section 2.3 – increase the injection volume is a vary common practice, do you think it is relevant to report in the text? In general, in the manuscript only the practice with best results is reported. I suggest moving the best chromatogram obtained as an example in the section 5.5 and, in my opinion, only the subsection 2.3.3 deserves to be mentioned.
Discussion – In my opinion this section is too long and in general it should not have subparagraphs. In fact, authors should discuss the results and their implications considering the existing literature.
Line 363 – 132 or 127 samples?
Line 373 – where are the results of heavy metal analyses
Section 5.5- I suggest adding the MS/MS spectrum of tetrodotoxin
Author Response
Reviewer 2:
The manuscript “First detection of tetrodotoxin in bivalves and 2 gastropods from the French mainland coasts” deal with the detection of tetrodotoxin in 4 samples (127 collected) of shellfishes from the French coasts. Tetrodotoxin were quantified in only one sample. Tetrodotoxin is commonly associated to the consumption of contaminated fish in China and Japan, but some intoxications have occurred also in Europe. This is the first detection of tetrodotoxin in France. The authors also optimized the analytical method for the detection of tetrodotoxin using HILIC-MS/MS.
I found this work interesting and it can be accepted after minor changes.
Answer: Thank you for your positive feedback.
Abstract - The abstract should be rewritten in order to be more concise and relevant. The first lines on tetrodotoxins are not necessary, especially because they detected only the founder compound of this group named tetrodotoxin.
Answer: As recommended, the abstract has been shortened. The first and the third lines were removed.
Only the founder compound was detected, however the analogues were also sought and occurrence data were also obtained for those compounds (< LOD). Therefore, we prefer to keep the term “tetrodotoxins” and not “tetrodotoxin” in the abstract.
Line 30- Please add references.
Answer: We agree and added the following reference [2]:
Silva, M.; Rodríguez, I.; Barreiro, A.; Kaufmann, M.; Neto, A.I.; Hassouani, M.; Sabour, B.; Alfonso, A.; Botana, L.M.; Vasconcelos, V. Tetrodotoxins occurrence in non-traditional vectors of the north atlantic waters (Portuguese maritime territory, and morocco coast). Toxins (Basel). 2019, 11, 1–17, doi:10.3390/toxins11060306.
Line 95- considering that the method validation reported in Appendix A is not complex and long, I suggest removing the appendix A and adding all the information in the main text.
Answer: as recommended, the appendix A was removed and introduced in the section 2.1.
Lines 124-131- is this necessary in the results section? I suggest reporting only the actual number of samples and the sampling sites.
Answer: We agree and moved these information in the section 5.2.
Section 2.3 – increase the injection volume is a vary common practice, do you think it is relevant to report in the text? In general, in the manuscript only the practice with best results is reported. I suggest moving the best chromatogram obtained as an example in the section 5.5 and, in my opinion, only the subsection 2.3.3 deserves to be mentioned.
Answer: We fully agree with you. It is clearly a better choice to keep only the practice with the best results reported. However, it is not possible in this case. Initially all samples were analysed with the TTX method. Close to the end of all the analysis, we observed that TTX was detected in few samples and at very low levels. The sensitive variant of the method was developed at that moment and was only applied to those few samples. To proceed as you suggest, all samples should have been re-analyzed with the sensitive variant of the method. This work would have been too time-consuming and expensive for us, so this strategy was not retained.
Discussion – In my opinion this section is too long and in general it should not have subparagraphs. In fact, authors should discuss the results and their implications considering the existing literature.
Answer: Subparagraphs were removed in section 3. as suggested by the reviewer. Regarding the length of the discussion, reviewer 1 highlighted in a comment that the discussion falls a little bit short, therefore it is difficult to answer to both reviewers on that point. The most relevant literature has already been cited in the discussion.
Line 363 – 132 or 127 samples?
Answer: This sentence was clarified (now in section 5.2). The sampling design included 132 bivalve mollusc samples, however, 127 were actually collected.
Line 373 – where are the results of heavy metal analyses
Answer: Heavy metals data were obtained by another team for another context, therefore, it was not possible to include those results in the present paper. They will be published later.
Section 5.5- I suggest adding the MS/MS spectrum of tetrodotoxin
Answer: Analysis were carried out in MRM and not in MS/MS mode, therefore it is not possible to include the MS/MSspectrum in the publication.